# Reproducibility Study of "Attack-Resilient Image Watermarking Using Stable Diffusion"

## Abstract

This paper presents a reproducibility study and robustness evaluation of the paper 'Attack-Resilient Image Watermarking Using Stable Diffusion' by Zhang et al. (2024), which proposes ZoDiac, a Stable Diffusion-based framework for attack-resilient image watermarking. While successfully replicating the original method's core claims—achieving >90% watermark detection rate (WDR) against diffusion-based regeneration attacks and across MS-COCO, DiffusionDB, and WikiArt datasets—we identify critical vulnerabilities under adversarial and geometrically asymmetric attack paradigms. Our extended analysis demonstrates that gradient-based adversarial perturbations reduce ZoDiac's WDR, a threat model absent in prior evaluations. We also investigate rotationally asymmetric attacks achieving WDR below 65%. Additionally, we explore a new loss function to mitigate these limitations. Despite these enhancements, composite attacks combining adversarial noise with other methods reduce WDR to near-zero, exposing vulnerabilities through multi-stage offensive pipelines. Our implementation can be found on Anonymous Github[1] .

## 1 Introduction

The rapid advancement of generative AI has heightened the need for strong image watermarking techniques to verify content authenticity and counter AI-generated forgeries Craver et al. (1998); Tirkel et al. (1994); Cox et al. (2007). Traditional watermarking methods, such as frequency-domain embeddings (Discrete Cosine Transform(DCT)(Bors & Pitas, 1996), Discrete Wavelet Transform(DWT)) and Least-Significant-Bit (LSB)(Wolfgang & Delp, 1996) manipulation, were designed to withstand standard distortions like JPEG compression and Gaussian noise. However, they struggle against modern diffusion-based regeneration attacks, which use latent-space purification to erase watermarks. Early neural approaches, including RivaGAN(Zhang et al., 2018) and StegaStamp(Tancik et al., 2020), improved resilience through adversarial training and spatial transformer networks. Yet, their pixel-space embeddings remain vulnerable to pipeline-aware attacks that exploit diffusion models iterative denoising to remove watermarks . This weakness arises because these methods operate in pixel space, making watermarks susceptible to latent-space purification.

Recent diffusion-based techniques aim to address this issue. Tree-Ring(Wen et al., 2023) watermarks encode concentric ring patterns into the initial noise vectors of synthetic images. By leveraging the deterministic inversion property of diffusion models, these watermarks can be recovered from generated outputs. Embedding patterns in the Fourier domain and utilizing the model's latent-space dynamics allows Tree-Ring to achieve rotational invariance and resist individual attacks. However, this approach only applies to synthetically generated images, leaving real-world content unprotected. Additionally, its reliance on isotropic ring patterns makes it vulnerable to asymmetric transformations. Its static design also lacks defenses against composite attacks that combine geometric distortions with purification-based techniques. StableSignature(Fernandez et al., 2023) fine-tune diffusion decoders to embed watermarks but require extensive training on large datasets, making them resource-intensive and reducing practicality.

---

[1]Link to Anonymous Github

ZoDiac addresses the aforementioned gaps through a novel framework that integrates pre-trained Stable Diffusion model with DDIM inversion to embed imperceptible watermarks. The method capitalizes on the bidirectional nature of diffusion models: it maps an input image into a latent vector via inversion, injects a ring-shaped watermark into the Fourier domain of this latent space, and reconstructs the image while preserving fidelity. Unlike pixel-space methods, ZoDiac operates in the latent space where diffusion models inherently resist purification attacks—iterative denoising during generation reinforces watermark persistence. ZoDiac explicitly aligns the injected watermark with its retrieved version in Fourier space, countering latent-space distortions caused by augmentations in the pixel-space. This contrasts with Tree-Ring's static synthetic-only embeddings, which lack such alignment mechanisms.

Our choice of the ZoDiac paper was based on the fact that the authors claim to have better results than the previous SOTA frameworks(e.g. Stegastamp(Tancik et al., 2020),SSL(Fernandez et al., 2022), CIN(Ma et al., 2022), etc.) without significant computational overheads such as in Stable Signature(Fernandez et al., 2023). Through this paper, we aim to reproduce the core claims of the original work, extending its evaluation with extended attack paradigms. We introduce adversarial and geometric attack scenarios, including directional blurring and chromatic distortions, to test the ZoDiac framework's resilience beyond the original evaluation scope. Additionally, we explore an augmented loss function incorporating L1 constraints in an attempt to enhance watermark detection under diverse attack conditions. Our experiments reveal critical vulnerabilities in the watermarking approach, particularly when facing composite attacks that combine adversarial noise with geometric transformations.

**Paper Outline**: Section 2 discusses the scope of reproducibility. Section 3 details the ZoDiac framework's latent-space watermarking methodology. Section 4 outlines our reproducibility efforts and validated claims. It describes our methodology, including datasets and evaluation metrics. Sections 5 and 6 discuss future directions and concludes with broader implications for watermarking in generative AI contexts.

## 2 Scope of Reproducibility

In this reproducibility study, we rigorously validate the fundamental claims of the ZoDiac framework while subjecting it to comprehensive stress tests across a range of attack paradigms. Our efforts are primarily directed toward verifying the four central claims outlined in the original work.:

- **Claim 1 :** ZoDiac demonstrates a watermark detection rate (WDR) exceeding 98 and a false positive rate (FPR) below 6.4 across MS-COCO, DiffusionDB, and WikiArt datasets, outperforming state-of-the-art watermarking methods.

- **Claim 2 :** ZoDiac remains resilient to diverse attack categories, including traditional attacks (such as JPEG compression and Gaussian blurring), Stable Diffusion-based regeneration attacks, where most other methods fail and rotational attacks to some extent.

- **Claim 3 :** Unlike prior methods like Tree-Ring or Stable Signature, ZoDiac can watermark both real-world and synthetic images without requiring retraining of the stable diffusion model, making it highly practical for diverse applications and real world deployment.

- **Claim 4 :** ZoDiac achieves imperceptible watermarks with image quality metrics such as SSIM $\geq 0.92$, ensuring minimal visual degradation while maintaining robustness against attacks. ZoDiac ensures a fair tradeoff between watermark detection and maintaining image quality.

## 3 Methodology

### 3.1 Description of Methods

The ZoDiac framework methodology comprises three key components: (1) latent-space vector initialization via DDIM inversion, (2) Fourier-domain watermark embedding for geometric resilience, and (3) adaptive image enhancement to balance detectability and visual fidelity. Below, we detail each component as described in the original paper.

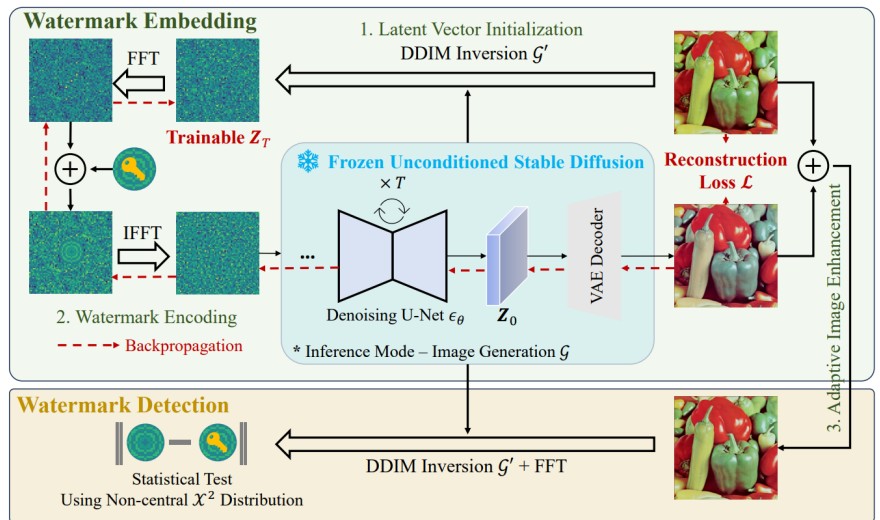

Figure 1: Overview of the ZoDiac framework, illustrating the watermark embedding and detection pipleines. Image taken from the original paper Zhang et al. (2024)

### 3.1.1 Latent Vector Initialization via DDIM Inversion

The process begins by mapping an input image $x_0$ to a latent vector $\mathbf{Z}_T$ using **DDIM inversion**:

$$\mathbf{Z}_T = \mathcal{G}'(x_0), \tag{1}$$

where $\mathcal{G}'$ denotes the inversion of the pre-trained Stable Diffusion model. The inversion adheres to the forward diffusion process:

$$\mathbf{Z}_{t-1} = \sqrt{\bar{\alpha}_{t-1}} \left( \frac{\mathbf{Z}_t - \sqrt{1 - \bar{\alpha}_t} \epsilon_\theta(\mathbf{Z}_t, t)}{\sqrt{\bar{\alpha}_t}} \right) + \sqrt{1 - \bar{\alpha}_{t-1}} \epsilon_\theta(\mathbf{Z}_t, t), \tag{2}$$

where $\bar{\alpha}_t$ controls the noise schedule and $\epsilon_\theta$ is the pre-trained denoiser. Initializing $\mathbf{Z}_T$ via inversion ensures faster convergence and preserves the structural integrity of the original image during watermark injection.

### 3.1.2 Fourier-Domain Watermark Encoding

ZoDiac injects a ring-shaped watermark $\mathbf{W}$ into the Fourier transform of $\mathbf{Z}_T$ to exploit rotational symmetry and frequency-domain resilience:

**Watermark Generation:** $\mathbf{W}$ is sampled from $\mathcal{CN}(0, 1)$ (complex Gaussian distribution), with elements equidistant from the center of latent vector being assigned identical values. A binary mask $\mathbf{M}$ localizes the watermark to low/mid frequencies:

$$\mathbf{M}_p = \begin{cases} 1 & \text{if } d(p, c) \leq d^* \\ 0 & \text{otherwise} \end{cases}, \tag{3}$$

where $d(p, c)$ is the Euclidean distance from coordinate $p$ to the latent center $c$, and $d^*$ is the mask radius.

**Watermark Injection:** The watermark is applied to the Fourier-transformed latent vector:

$$\mathcal{F}(\mathbf{Z}_T)[ic, :, :] = (1 - \mathbf{M}) \odot \mathcal{F}(\mathbf{Z}_T)[ic, :, :] + \mathbf{M} \odot \mathbf{W}, \tag{4}$$

where $\mathcal{F}(Z_T) \in \mathbb{C}^{ch \times w \times h}$ is the Fourier transform of latent vector, $ic$ is the target watermark injection channel and $\odot$ represents element wise product. We denote the latent vector after watermarking as $\mathbf{Z}_T \oplus \mathbf{W}$.

**Latent Optimization:** The watermarked latent $\mathbf{Z}_T \oplus \mathbf{W}$ is optimized via gradient descent to minimize a multi-term reconstruction loss:

$$\mathcal{L} = \underbrace{\|\hat{x}_0 - x_0\|_2}_{\text{L2}} + \lambda_s \mathcal{L}_{\text{SSIM}} + \lambda_p \mathcal{L}_{\text{Watson-VGG}}, \tag{5}$$

where $\mathcal{L}_{\text{SSIM}}$ is SSIM loss(Zhao et al., 2017) which preserves structural similarity , and $\mathcal{L}_{\text{Watson-VGG}}$ corresponds to the Watson-VGG perceptual loss(Czolbe et al., 2021) enforcing perceptual fidelity using features extracted by pre-trained VGG network.

### 3.1.3 Adaptive Image Enhancement

To preserve visual quality while maintaining watermark robustness, ZoDiac employs an adaptive blending mechanism between the watermarked image $\hat{x}_0$ and original $x_0$ through the parameterized operation:

$$\bar{x}_0 = \hat{x}_0 + \gamma(x_0 - \hat{x}_0), \tag{6}$$

where the blending coefficient $\gamma \in [0, 1]$ is optimized via binary search to satisfy structural fidelity constraints:

$$\min \gamma \quad \text{s.t.} \quad \text{SSIM}(\bar{x}_0, x_0) \geq s^*. \tag{7}$$

This formulation explicitly negotiates the trade-off between imperceptibility and watermark detectability (measured via WDR). The dynamic adaptation mechanism enables automatic quality control across diverse image characteristics - a critical improvement over rigid approaches like StegaStamp and RivaGAN, which exhibit quality degradation under latent-space perturbations. By construction, the blending process preserves high-frequency watermark components while suppressing low-frequency artifacts, ensuring both visual fidelity and attack resilience.

### 3.1.4 Watermark Detection via Statistical Testing

Watermark detection involves performing a null hypothesis test on the presence of $\mathbf{W}$ in binary mask $\mathbf{y}$ of reconstructed image latent $\mathbf{Z}_T$:

1. **DDIM Inversion:** Reconstruct the latent $\mathbf{Z}'_T = \mathcal{G}'(x'_0)$ from the (potentially attacked) image $x'_0$.

2. **Fourier Extraction:** Compute the watermark binary mask $\mathbf{y} = \mathcal{F}(\mathbf{Z}'_T)[-1, :, :]$.

3. **Non-Central Chi-Squared Test:**

   (a) Null hypothesis $H_0 : \mathbf{y} \sim \mathcal{N}(0, \sigma^2 \mathbf{I})$.

   (b) Test statistic: $\eta = \frac{1}{\sigma^2} \sum (\mathbf{M} \odot \mathbf{W} - \mathbf{M} \odot \mathbf{y})^2$. Under $H_0$, $\eta$ follows a non-central chi-squared distribution(Patnaik, 1949)

   (c) Reject $H_0$ if $(1 - p) > p^*$, where $p$ is obtained from the $\chi^2$ CDF with $\sum \mathbf{M}$ degrees of freedom. Here, $(1 - p)$ represents the likelihood of watermark presence and $p^*$ is the set threshold; hence, an image is deemed watermarked if $(1 - p) > p^*$.

## 3.2 Evaluation Metrics

To comprehensively assess ZoDiac's performance, we use four quantitative metrics spanning detection robustness, image quality and attack resilience. These metrics align with established benchmarks in the field.

**Detection Robustness:** The **Watermark Detection Rate (WDR)** is calculated as $\text{WDR} = \frac{\text{TP}}{\text{TP} + \text{FN}}$, measuring the proportion of watermarked images correctly identified under attack. The **False Positive Rate (FPR)** is given by $\text{FPR} = \frac{\text{FP}}{\text{FP} + \text{TN}}$.

**Image Quality Preservation:** This is measured by evaluating

- **Peak Signal-to-Noise Ratio (PSNR):** Defined as

$$\text{PSNR}(x, \bar{x}) = -10 \log_{10}(\text{MSE}(x, \bar{x}))$$

- **Structural Similarity Index (SSIM):** (Zhao et al., 2017) Enforced as SSIM $\geq 0.92$ through adaptive blending.

## 3.3 Experimental Setup

We obtained the code from the Github[2] repository provided by the original authors and greatly appreciate that their well-structured code was easy to understand and modify. While the original codebase contained the core functionality, though somewhat helpful, it was primarily structured as a demonstration notebook rather than a comprehensive framework for experimental validation. Consequently, one of our key contributions was modifying the original repository to include well-organized and generalizable scripts for all the experiments presented in our paper, as well as for practical applications. All of our code is available here: Anonymous Github[3].

## 3.4 Datasets

**Datasets:** ZoDiac is evaluated across three domains to assess generalizability: **MS-COCO**(Lin et al., 2015) (Real-world photographs, 80,000+ images) to test robustness on natural scenes with complex textures and lighting. A subset of 500 images is randomly sampled from the validation set. **DiffusionDB**(Wang et al., 2023) (AI-generated images, 1.6M+ images) Using a subset of 500 images generated with diverse text prompts. **WikiArt**(Phillips & Mackintosh, 2011) (Artistic works, 250,000+ paintings across 195 styles) validates performance on non-photographic content with unique color palettes and brushstrokes, with a subset of 500 images.

We use an equal number of randomly sampled images from each dataset for all our experiments unless specified otherwise. We use the same datasets as the original paper as we deem the relevance and diversity brought on by these datasets in terms of real and ai-generated images, artistic styles and variety of lighting and textures sufficient for all practical purposes.

## 3.5 Computational Requirements

We evaluated ZoDiac's demands on consumer-grade GPUs for reproducibility. The watermarking pipeline (latent vector initialization and adaptive enhancement) was run on an NVIDIA P100 (16GB VRAM), taking about 295–320 seconds per image (50 denoising steps, 100 optimization iterations). Adversarial attacks (PGD, 50 steps) were executed on an NVIDIA A6000 (48GB VRAM), requiring 820–950 seconds per image due to increased memory needs; these attacks are also feasible on 16GB GPUs at reduced speed. Overall, robustness testing (including DDIM inversion and statistical test) averaged 2 minutes per image on the P100.

| Script | Time (in hours) | Kgs of $CO_2$ |
|---|---|---|
| Basic Watermarking | 13.6 | 2.38 |
| All Attacks (except adv) | 5.3 | 0.92 |
| Adversarial Attacks | 15.8 | 2.77 |

Table 1: GPU usage for a batch of 50 images on two different scripts for 50 denoising steps, 100 training iterations, and 50 steps of PGD on a single P100. Quantity of $CO_2$ estimated using the Machine Learning Impact calculator(Lacoste et al., 2019).

---

[2]Link to original paper's GitHub
[3]Link to Anonymous Github

| Detection Threshold | FPR ↓ | Watermark Detection Rate (WDR) ↑ | | | | | | | | | | | |
|---|---|---|---|---|---|---|---|---|---|---|---|---|---|
| | | Pre | Bright. | Cont. | JPEG | G-Noise | G-Blur | BM3D | Bmshj | Cheng | Zhao | Rot. | All |
| | | | | | | MS-COCO | | | | | | | |
| 0.90 | 0.058 | 1.000 | 1.000 | 1.000 | 0.992 | 1.000 | 1.000 | 1.000 | 1.000 | 0.960 | 0.980 | 0.516 | 0.080 |
| 0.95 | 0.014 | 1.000 | 0.996 | 1.000 | 0.988 | 0.998 | 1.000 | 1.000 | 0.920 | 0.958 | 0.974 | 0.316 | 0.080 |
| 0.99 | 0.004 | 0.996 | 0.960 | 0.960 | 0.952 | 0.984 | 0.960 | 0.952 | 0.910 | 0.930 | 0.938 | 0.106 | 0.000 |
| | | | | | | DiffusionDB | | | | | | | |
| 0.90 | 0.052 | 1.000 | 0.998 | 0.998 | 0.988 | 0.978 | 0.984 | 0.960 | 0.980 | 0.990 | 0.956 | 0.530 | 0.070 |
| 0.95 | 0.012 | 1.000 | 0.996 | 0.996 | 0.980 | 0.974 | 0.976 | 0.956 | 0.972 | 0.980 | 0.906 | 0.316 | 0.000 |
| 0.99 | 0.002 | 0.996 | 0.974 | 0.960 | 0.964 | 0.950 | 0.960 | 0.950 | 0.950 | 0.964 | 0.860 | 0.080 | 0.000 |
| | | | | | | WikiArt | | | | | | | |
| 0.90 | 0.058 | 1.000 | 0.980 | 0.980 | 0.980 | 0.980 | 1.000 | 0.980 | 0.976 | 0.960 | 0.980 | 0.428 | 0.040 |
| 0.95 | 0.018 | 1.000 | 0.980 | 0.980 | 0.984 | 0.980 | 1.000 | 0.980 | 0.964 | 0.960 | 0.960 | 0.290 | 0.020 |
| 0.99 | 0.002 | 1.000 | 0.974 | 0.964 | 0.972 | 0.944 | 0.966 | 0.960 | 0.958 | 0.942 | 0.912 | 0.080 | 0.000 |

Table 2: Effects of varying detection thresholds $p^* \in \{0.90, 0.95, 0.99\}$ on watermark detection rate (WDR) and false positive rate (FPR) for all attacks. WDR measured on watermarked images with SSIM threshold $s^* = 0.92$.

# 4    Results

We present our results in two subsections, first reproducing the results from the original paper and then presenting the extended results from experiments beyond the original paper.

## 4.1    Results From the Original Paper

### 4.1.1    Verifying Claim 1: Claimed WDR

We evaluated WDR/FPR on 500-image subset from each dataset. We tested with different detection thresholds present our findings in Table 2. As claimed by the original paper, ZoDiac demonstrates superior robustness across diverse datasets (MS-COCO, DiffusionDB, WikiArt) and attack scenarios: adjustments in brightness or contrast,JPEG compression,Image rotation,Gaussian noise,Gaussian blur,BM3D denoising(Dabov et al., 2007),Bmshj(Ballé et al., 2018),Cheng(Cheng et al., 2020),Zhao(Zhao et al., 2023); achieving >98% Watermark Detection Rate (WDR). Minor discrepancies (<5%) fall within acceptable bounds, reinforcing the original claims' validity. Thus, this claim is verified and there is no further disussion or reason to suggest otherwise.

### 4.1.2    Verifying Claim 2: Attack Resilience

As demonstrated in our experiments, we successfully reproduced the performance of ZoDiac on all metrics explored in the original paper. However, after identifying limitations inherent to the SSIM metric, we devised novel attack classes designed to exploit these weaknesses. While the authors' claims regarding robustness against the originally evaluated attacks are confirmed as shown in Table 2, our findings highlight that the general robustness of ZoDiac can be compromised under extended attack scenarios as discussed in 4.2.

### 4.1.3    Verifying Claim 3: Deployment Practicality

Watermark injection required 295–320 seconds/image on an NVIDIA P100 GPU (16GB VRAM), aligning with the original paper's reported 255.9s/image on an RTX8000, with minor latency variations attributable to GPU architecture differences. While this per-image latency poses challenges for real-time deployment, ZoDiac's elimination of upfront training costs starkly contrasts with alternatives like Stable Signature, which requires >100 GPU hours to fine-tune the diffusion decoder on a 100K-image dataset.

While ZoDiac's per-image latency exceeds traditional methods like DwtDct(Bloom et al., 1999) (<10s/image), its robustness justifies the trade-off in non-real-time scenarios (e.g., archival systems). Batch processing 100 images parallelized across 4×P100 GPUs reduces effective latency to <2 hours, comparable to Stable Signature's training duration for a single model iteration.

ZoDiac's zero-shot design, provides a viable solution for watermarking existing content without costly re-training. Despite higher per-image latency than non-diffusion methods, its elimination of upfront training (unlike Stable Signature) and dual real/synthetic compatibility (unlike Tree-Ring) make it practical for enterprise-scale deployment. Hence, this claim is also justified.

### 4.1.4 Verifying Claim 4: Image Quality

Our experiments corroborate ZoDiac's ability to preserve visual fidelity while embedding watermarks, achieving SSIM > 0.91 across all evaluated datasets (MS-COCO, DiffusionDB, WikiArt). These results closely align with the original paper's reported values, with minor variations attributable to stochastic initialization during latent optimization. The inclusion of SSIM and perceptual losses in ZoDiac's training objective inherently enforces fidelity, ensuring watermarked images remain visually indistinguishable from originals.

While ZoDiac prioritizes imperceptibility, ablation studies reveal a predictable trade-off: stricter SSIM thresholds (e.g., SSIM > 0.95) reduce watermark robustness by nearly 80% WDR under composite attacks. However, the original paper's recommended threshold (SSIM = 0.92) balances this trade-off effectively, as reproduced in our study.

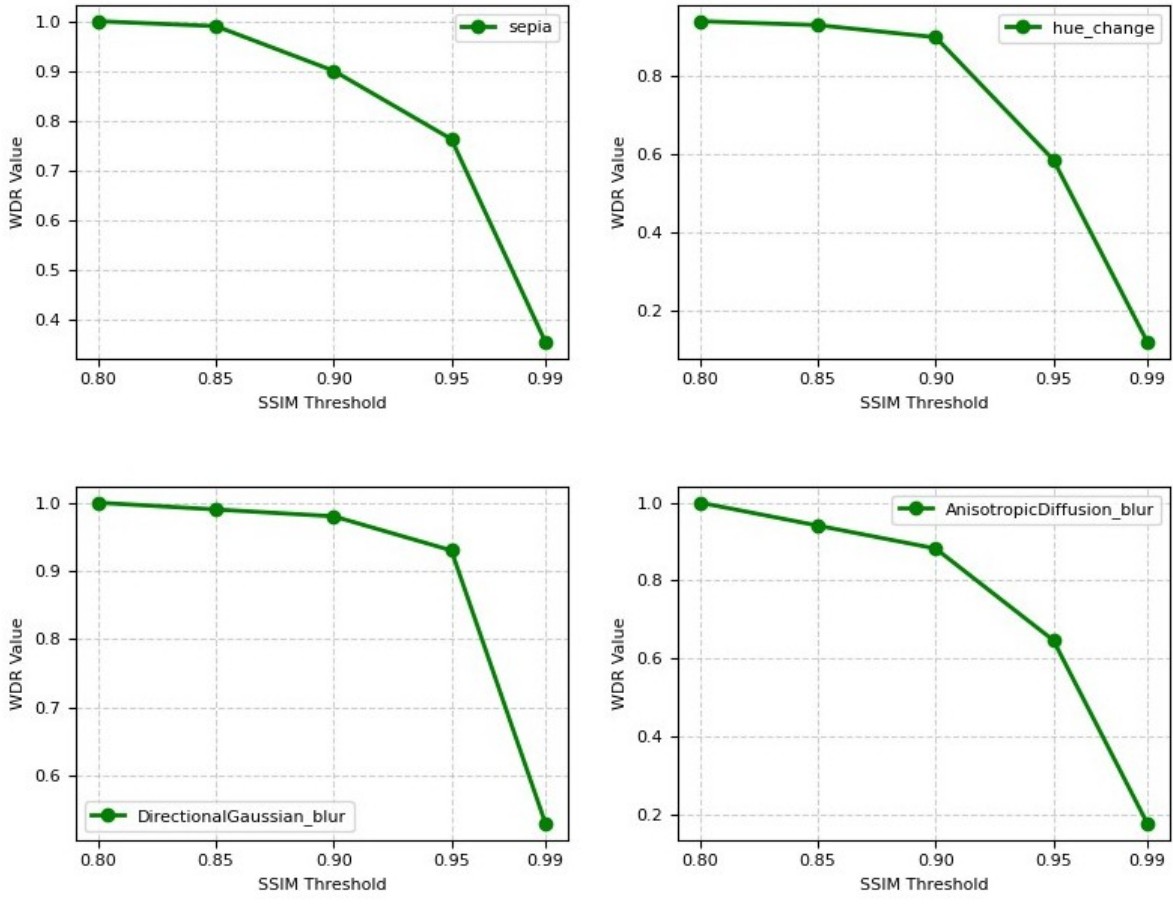

Figure 2: Variation of WDR at $p^* = 0.9$ with different SSIM thresholds for Sepia, Huechange, Directional-GaussianBlur, AnisotropicDiffusionBur attackers.

The reproduced results validate the claim, confirming ZoDiac's capacity to embed watermarks imperceptibly while preserving image quality. Thus this claim is also verified.

## 4.2 Experiments Beyond the Original Paper

In this section we present additional attacks, some of which have been well established in Kutter & Petitcolas (1999). Furthermore, we explore augmenting the loss function. Finally, we simulate adversarial attacks by a knowledgeable adversary (Choubassi & Moulin, 2005; Comesaña et al., 2006).

### 4.2.1 Directional Blurring Attacks

The original paper evaluated robustness against isotropic Gaussian blurs but did not address directional blurring attacks—geometrically asymmetric perturbations that exploit rotational dependencies in Fourier-domain watermark embeddings. Our experiments extend ZoDiac's evaluation to directional blurring kernels, revealing latent-space vulnerabilities tied to circular symmetry assumptions of the framework.

ZoDiac's radial Fourier mask $M$ assumes invariance under rotational transformations. Directional blurs violate this assumption, perturbing latent vectors $Z_T$ via:

$$\mathcal{F}(Z_T)_{rot} = R_\theta(\mathcal{F}(Z_T)) \tag{8}$$

where $R_\theta$ denotes rotation by $\theta$. This disrupts the consistency of the DDIM inversion, as transformed latents map to distinct $x_0$ reconstructions. Therefore, we hypothesize that attacking through directional blurring schemes will produce significant change in the WDR of the framework.

The implementation of this experiment considers WDR with baseline blurring (Gaussian Blurring) and two directinal blurring techniques- Anisotropic Blurring and Motion Blurring, with the following parameter settings:

| Attack Type | Description |
| --- | --- |
| Anisotropic Blur | 45°-aligned Gaussian kernels ($\sigma = 3$) applied along non-radial axes. |
| Motion Blur | Linear kernels (length $= 15$px) at 30° angles, simulating camera motion. |
| Baseline | Isotropic Gaussian blur ($\sigma = 3$) as used in the original study. |

Table 3: Descriptions of Blur Attacks and Baseline

Table 4 displays some of the results obtained while the complete lists is given in A.1.

### 4.2.2 Changing Color Hue

While ZoDiac demonstrates robustness against diffusion and noise-based attacks, its reliance on SSIM for quality control introduces vulnerabilities to chromatic distortions. Structural Similarity Index (SSIM) emphasizes luminance and structural fidelity but exhibits limited sensitivity to hue shifts—a critical gap given real-world attack vectors like selective color grading or adversarial hue perturbations. Our experiments evaluate ZoDiac under systematic hue perturbations, revealing SSIM's failure to capture perceptually significant color distortions that degrade watermark alignment.

Continuing work along these lines , we analyze two more methods - Color Quantization and Sepia filter. Analyzing the attached graphic, Color Quantization and Sepia tone application can be effective at disrupting the ZoDiac watermark. Color Quantization reduces the number of distinct colors in an image, consolidating similar hues into a limited palette. This process compromises high-frequency details and fine gradients, creating block-like artifacts especially notable in smoother regions. Sepia toning, conversely, is a form of color palette reduction that maps original colors to shades of brown, creating a monochromatic aesthetic. While SSIM aims to capture structural similarity across images, it remains relatively insensitive to broad color palette modifications. For ZoDiac, these attacks can induce misregistrations by distorting relationships between chromatic channels, leading to phase corruptions in the embedded Fourier domain and disrupting the DDIM inversion process during watermark detection; furthermore both techniques can introduce signal loss with can increase false negative scores. These vulnerabilities highlight that ZoDiac requires additional strategies for watermarking under perceptually relevant color space manipulations.

| | Gaussian | Anisotropic | Sepia Filter | Color Quantization | Hue Change |
|---|---|---|---|---|---|
| **WDR** | 0.994 | 0.646 | 0.902 | 0.807 | 0.773 |

Table 4: Watermark Detection Rate (WDR) for various attack types: blurring kernels and color filters.

### 4.2.3 Rotational and Geometric Attacks

In this section, we examine rotation-based and lateral inversion attacks that exploit the geometric dependencies inherent to ZoDiac's Fourier-domain watermark embeddings. ZoDiac embeds watermarks as concentric rings in Fourier space, relying on their radial symmetry to withstand common distortions. However, rotation-based attacks where the image is rotated by arbitrary angles disrupt this symmetry by misaligning the Fourier mask relative to the original watermark pattern. This misalignment introduces phase shifts that degrade the statistical detection test, resulting in significant drops in the watermark detection rate (WDR), as demonstrated by the marked fluctuations observed with varying rotation angles.(Figure 3)

Similarly, lateral inversion (i.e., horizontal flipping) alters the spatial configuration of the watermark in a manner not anticipated by ZoDiac's detection framework. This inversion modifies the orientation of the latent-space representation, further exacerbating misalignment during DDIM inversion and Fourier-based detection. Both rotation and inversion attacks leverage the absence of inherent geometric invariance in ZoDiac's watermarking scheme, rendering it vulnerable to transformations that change spatial relationships without causing perceptible image quality loss.(Results depicted in Table 7)

These observations underscore a key limitation of ZoDiac's design—its exclusive reliance on circularly symmetric Fourier embeddings without additional mechanisms for geometric correction. To mitigate these vulnerabilities, future work could explore the integration of rotation-invariant embeddings or automated realignment techniques. Notably, the combination of rotation and inversion attacks can reduce the WDR to nearly zero, effectively evading watermark detection while preserving image fidelity. Detailed results are displayed in Appendix A.1

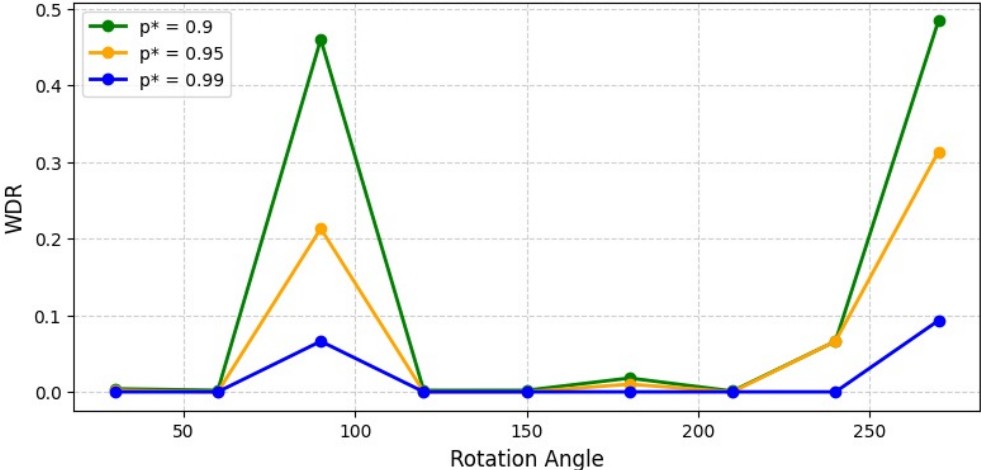

Figure 3: Variation of WDR with change in Rotation Angle at different Detection Threshold ($p^*$)

### 4.2.4 Augmenting the Loss Function

The original ZoDiac framework employs the reconstruction loss:

$$L = \|\hat{x}_0 - x_0\|^2 + \lambda_s L_{\text{SSIM}} + \lambda_p L_{\text{Watson-VGG}} \tag{9}$$

to preserve image fidelity while optimizing latent vectors. While effective in basic reconstruction scenarios, this formulation lacks explicit constraints on watermark alignment between injected **W** and reconstructed **Ŵ** patterns, potentially allowing latent-space misregistrations under complex attack scenarios.

Following established principles in the watermarking domain, we integrate direct watermark fidelity into the loss function by incorporating an $L_1$ distance term. This methodology aligns with conventional watermarking techniques that leverage a combination of $L_1$ and $L_2$ constraints to concurrently uphold perceptual quality and enhance watermark robustness(Zhu et al., 2018; Volpi & Tuia, 2018; Wan et al., 2019; Zhang et al., 2019; Li et al., 2019).

The authors of the original paper have not discussed this loss in the paper but have provided this as an option in the code implementation. Therefore, our extension includes the $L_1$ distance term to attempt enforcement of direct watermark fidelity and improve representational stability.

**Mechanistic Analysis**

The $L_1$ term directly minimizes the Manhattan distance between $W$ and $\hat{W}$ in the complex Fourier domain:

$$\|W - \hat{W}\|_1 = \sum_p \left| \text{Re}(W_p - \hat{W}_p) \right| + \left| \text{Im}(W_p - \hat{W}_p) \right| \tag{10}$$

This enforces phase consistency in concentric rings, reducing misalignment under attack-induced perturbations. Our augmented loss thus becomes:

$$L = \underbrace{\|\hat{x}_0 - x_0\|^2}_{L_2} + \lambda_s L_{\text{SSIM}} + \lambda_p L_{\text{Watson-VGG}} + \lambda_{L1} \|W - \hat{W}\|_1 \tag{11}$$

where $\lambda_{L1} = 0.1$ balances watermark alignment and visual fidelity.

Empirical evaluation demonstrates that this approach introduces complex trade-offs between different attack resilience mechanisms, highlighting the fundamental challenge in watermarking systems that simultaneously optimize for multiple conflicting objectives, i.e., imperceptibility, robustness, and capacity. The $L_1$ constraint notably increased resilience to most attack methods, particularly enhancing detection rates for color manipulation attacks (Sepia filters and Color Quantization showing 5.0% and 9.3% improvements respectively). However, some minor performance variations were observed across different attack vectors, with some showing practically no change in WDR, and some infact achieving better attack success. The complete analysis of performance changes resulting from our loss function adjustment is provided in Appendix B.

|  | Gaussian | Anisotropic | Sepia Filter | Color Quantization |
|---|---|---|---|---|
| **WDR** | 0.975 | 0.675 | 0.950 | 0.900 |

Table 5: Watermark Detection Rate (WDR) for various attack types with loss changed: blurring kernels and color filters.

### 4.2.5 Adversarial Attack by a knowledgeable adversary

ZoDiac's original robustness claims focus on purification and conventional attacks but omit adversarial perturbations. We extend its threat model to include white-box gradient-based attacks where adversaries perturb the watermarked image to degrade watermark detection. The attacker's objective is to maximize the deviation between the adversarial latent's Fourier components and the original watermark region, constrained by an $\ell_\infty$-norm bound ($\epsilon = 0.05$). This targets the statistical detection mechanism in ZoDiac's Fourier space.

**Attack Methodology**

We formulate the adversarial optimization using Projected Gradient Descent (PGD)(Madry et al., 2019) over 50 iterations:

$$Z_T^{adv} = \text{proj}_\epsilon \left( Z_T^{adv} + \alpha \cdot \text{sign} \left( \nabla_{Z_T} L_{adv} \right) \right), \tag{12}$$

where the adversarial loss function is defined as:

$$L_{adv} = \left\| M \odot \left( \mathcal{F}(Z_T^{adv}) - \mathcal{F}(Z_T) \right) \right\|_1. \tag{13}$$

By perturbing the latent vector's Fourier components within this region, the attack disrupts the chi-squared statistical test's assumptions, reducing detection confidence. Under standalone adversarial attacks ($\epsilon = 0.05$), ZoDiac's watermark detection rate (WDR) drops to 75.74%, (Table 6).

Furthermore combining adversarial noise with or geometric attacks amplifies robustness degradation. The synergy arises because adversarial perturbations destabilize the latent vector's Fourier structure, while subsequent attacks exploit residual vulnerabilities. Some of these results are tabulated in Table 6 while a complete list is provided in Appendix C.

**Infeasibility of Adversarial Training**

Adversarial training—fine-tuning ZoDiac on adversarially perturbed latents—is computationally prohibitive. Each PGD iteration requires 820–950 seconds/image on an NVIDIA A6000 GPU. A standard 10-iteration training protocol would demand over 14 hours per image, translating to more than 21,000 GPU hours for a 1,500-image evaluation set. This stems from backpropagation through the full denoising process during latent optimization, which cannot be parallelized due to DDIM's sequential nature.

Although defenses like randomized thresholds or Fourier-space noise injection could mitigate attacks but might also inadvertently damage watermark detection. Iterative adversarial training remains impractical. ZoDiac's reliance on pre-trained stable diffusion exacerbates this limitation, as retraining the backbone model would negate its zero-shot advantage.

**Implications**

ZoDiac's latent-space watermarking, while robust to purification, is vulnerable to coordinated adversarial-geometric attacks. Future work should explore lightweight defenses, such as Fourier-domain noise augmentation during watermark injection, to disrupt gradient-based exploits without retraining.

| Attack Type | WDR |
|---|---|
| Adversarial ($\epsilon = 0.05$) | 0.755 |
| Adversarial + DiffAttacker60 | 0.491 |
| Adversarial + Rotation(180) | 0.018 |

Table 6: WDR under standalone adversarial attack and composition of adversarial attack with other attacks.

## 5 Discussion

### 5.1 Limitations & Challenges

**White-Box Vulnerability:** Our adversarial attack analysis assumed complete knowledge of the watermark parameters. While higher PGD budgets can introduce visible artifacts in the generated images, our focus was on demonstrating the feasibility of incorporating adversarial attacks into the evaluation framework rather than an exhaustive study of partial-knowledge scenarios. The attacks could potentially be adapted for settings with incomplete information, though this remains outside our current scope.

**Computational Overhead:** The latent optimization process (~300s/image) and adversarial attack computation present deployment challenges, particularly for large-scale applications. While these computational

requirements are reasonable compared to alternative methods, they may constrain practical implementation when processing numerous images.

## 5.2 Future Directions

**Certified Adversarial Robustness:** Integrating some defense mechanisms to mitigate adversarial attacks without compromising WDR.

**Message Encoding:** Extending ZoDiac's zero-bit watermarking framework to multi-bit watermarks for provenance tracking.

**Auto Correction Defense:** Integrating auto correction defense to help defend against rotation- and lateral-inversion-based attacks is absolutely crucial to recover WDR while also maintaining the FPR through extensive training / fine-tuning / parameter selection.

**Efficient Implementation:** While effective against various attacks including diffusion-reconstruction ones, it requires ~300 seconds per image, making real-time applications challenging.

**Better Reconstruction Loss Term:** Integration of a better reconstruction loss term with the extracted watermark in the loss function, which may consistently improve the WDR across benchmarks, as opposed to the $L_1$ loss.

## 5.3 Broader Implications

This work underscores the dual role of diffusion models as both attackers and defenders in the watermarking arms race. By open-sourcing attack implementations and training protocols, we invite the research community to adopt standardized composite benchmarks (e.g., adversarial+geometric+regeneration) and prioritize defense-in-depth strategies. ZoDiac's success demonstrates that generative AI, often viewed as a threat, can be repurposed as a guardian of digital authenticity—a critical step toward ethical AI deployment.

# 6 Conclusion

Our reproducibility study validates ZoDiac's core premise of latent-space watermarking via a pre-trained Stable Diffusion model, achieving robust detection rates (WDR > 98%) and high perceptual quality (SSIM > 0.91) under conventional attack scenarios. Nonetheless, extended evaluations reveal significant vulnerabilities when subjected to directional blurring, adversarial, rotational and composite adversarial-geometric attacks, which severely undermine detection efficacy. The incorporation of an augmented loss function with an $L_1$ fidelity term yields partial improvements, particularly against chromatic distortions, yet it introduces trade-offs that affect performance across diverse attack vectors. These results underscore the need for further research into rotation-invariant embeddings and efficient adversarial defenses that can harmonize imperceptibility, robustness, and computational practicality.

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

## A  Extended Attack Results

### A.1  Comprehensive Attack Performance and Configurations

Here is the list of configurations of all the attacks we used in this paper. Attacks missing from this list either did not have any hyperparameters or the settings were self explanatory.

**Hyperparameter Selections**

- **DiffWM Attack**: Noise step = 60

- **Cheng2020-Anchor Compression**: Quality = 3

- **BMSHJ2018-Factorized Compression**: Quality = 3

- **JPEG Compression**: Quality = 1

- **Rotation Attack**: Degree = 30

- **Brightness Attack**: Brightness = 0.5

- **Contrast Attack**: Contrast = 0.5

- **Vibrancy Attack**: Vibrancy = 1.25

- **Gaussian Noise Attack**: Standard Deviation = 0.05

- **Gaussian Blur Attack**: Kernel Size = 5, $\sigma = 1$

- **Anisotropic Diffusion Blur Attack**: Num Iter = 15, $\delta_t = 0.14$, $\kappa = 50$

- **Directional Gaussian Blur Attack**: Kernel Size = 15, $\sigma = 5$, Angle = 45

- **Sharpening Attack**: Factor = 2.0

- **Salt and Pepper Noise Attack**: Amount = 0.1

- **Hue Change Attack**: Factor = 0.1

- **Elastic Deformation Attack**: $\alpha = 1000$, $\sigma = 50$

- **RGB to HSV Attack**: H-Shift = 0.1, S-Scale = 1.2, V-Scale = 1.1

- **Color Balance Attack**: R-Scale = 1.2, G-Scale = 1.0, B-Scale = 0.8

- **Gamma Correction Attack**: $\gamma = 1.5$

- **Log Transform Attack**: c = 1

- **Color Jitter Attack**: Brightness = 0.2, Contrast = 0.2, Saturation = 0.2, Hue = 0.1

- **Color Quantization Attack**: Number of Colors = 32

- **Posterization Attack**: Levels = 4

**Results:**

| Attacker | 30° | 60° | 90° | 120° | 150° | 180° | 210° | 240° | 270° | Lat. Rot. | Lat. Inversion |
|---|---|---|---|---|---|---|---|---|---|---|---|
| WDR | 0.004 | 0.002 | 0.460 | 0.002 | 0.002 | 0.018 | 0.001 | 0.002 | 0.484 | 0.007 | 0.837 |

Table 7: Variation of WDR with the rotation angle and lateral inversion or a combination (Lat. Rot. means Lateral Inversion + Rotation by 180°)

| Attacker Name | WDR at 0.9 | WDR at 0.95 | WDR at 0.99 |
|---|---|---|---|
| DiffWM Attacker | 0.933 | 0.873 | 0.727 |
| Black and White Attack | 0.913 | 0.853 | 0.787 |
| Lateral Inversion Attack | 0.960 | 0.940 | 0.853 |
| Sharpening Attack | 0.987 | 0.987 | 0.960 |
| Salt and Pepper Noise Attack | 0.833 | 0.673 | 0.427 |
| Hue Change Attack (0.3) | 0.773 | 0.673 | 0.500 |
| Elastic Deformation Attack | 0.967 | 0.967 | 0.927 |
| RGB to HSV Attack | 0.980 | 0.953 | 0.920 |
| Color Balance Attack | 0.993 | 0.973 | 0.940 |
| Gamma Correction Attack | 0.987 | 0.980 | 0.947 |
| Histogram Equalization Attack | 0.993 | 0.973 | 0.933 |
| Log Transform Attack | 1.000 | 1.000 | 0.940 |
| Color Jitter Attack | 0.960 | 0.933 | 0.887 |
| Color Quantization Attack | 0.807 | 0.726 | 0.600 |
| Sepia Attack | 0.900 | 0.840 | 0.773 |
| Posterization Attack | 0.960 | 0.940 | 0.907 |
| Directional Gaussian Blur Attack | 0.947 | 0.933 | 0.853 |

Table 8: WDR of all the attacks other than the Rotation based attacks at different $p^*$ values.

Note: Adversarial attack configurations discussed separately in Section C. Composite attacks ('all', 'all_norot') use parameter unions from above.

### A.2 Hyperparameter Sensitivity Analysis for Chromatic Distortion Attacks

We extend the original paper's brightness ($\delta_{bright}$) and contrast ($\gamma$) parameter analysis with additional hue variation studies ($\Delta h$). The attack transformations are:

- **Brightness**: $\mathcal{I}' = \text{clip}(\mathcal{I} \cdot \delta_{bright})$ for $\delta_{bright} \in \{0.2, 0.4, 0.5, 0.6, 0.8, 1.25, 1.5\}$

- **Contrast**: $\mathcal{I}' = \mu + \gamma(\mathcal{I} - \mu)$ for $\gamma \in \{0.2, 0.4, 0.5, 0.6, 0.8, 1.25, 1.5\}$

- **Hue(Our Extension)**: $\mathcal{I}'_{HSV} = \mathcal{I}_{HSV} + \Delta h$ for $\Delta h \in \{0.1, 0.3, 0.5, 0.7\}$ in HSV space

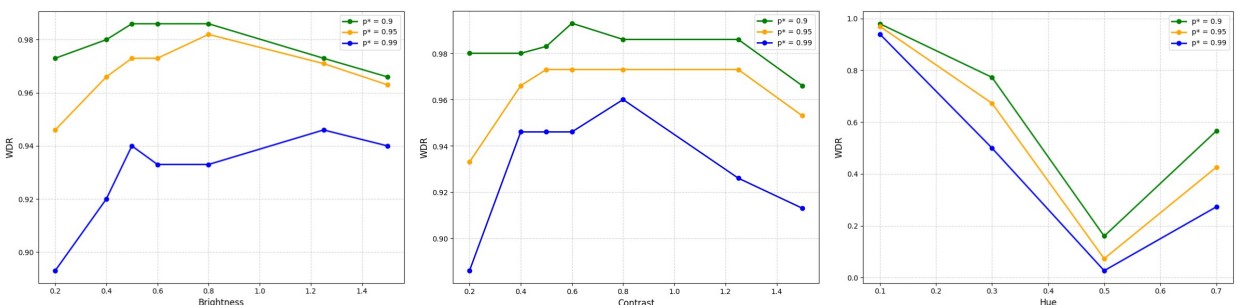

Figure 4: Variation of attack success rates for brightness, contrast (original paper's parameters) and hue variations (our extension) with change in their respective parameters ($\delta_{bright}, \gamma, \Delta h$).

## B Impact of Reconstruction Loss Term

We analyze the effects of modifying loss function as described in Section on our complete attack set.

| Attacker | 30° | 60° | 90° | 120° | 150° | 180° | 210° | 240° | 270° | Lat. Rot. | Lat. Inversion |
|----------|------|------|------|------|------|------|------|------|------|-----------|----------------|
| Original | 0.004 | 0.002 | 0.460 | 0.002 | 0.002 | 0.018 | 0.001 | 0.002 | 0.484 | 0.007 | 0.837 |
| Modified | 0.004 | 0.002 | 0.550 | 0.002 | 0.000 | 0.020 | 0.000 | 0.001 | 0.500 | 0.006 | 0.950 |

Table 9: Rotation-Based Attacks with original and modified loss for comparison

| Excluded Attacker Names | Original | Modified |
|-------------------------|----------|----------|
| Brightness 0.5 | 0.987 | 0.975 |
| Contrast 0.5 | 0.973 | 0.975 |
| JPEG Compression | 0.687 | 0.775 |
| Gaussian Noise Attack | 0.993 | 0.975 |
| Gaussian Blur Attack | 0.993 | 0.975 |
| BM3D Attack | 0.993 | 1.000 |
| BMSHJ2018-Factorized Compression | 0.960 | 0.950 |
| Cheng2020-Anchor Compression | 0.973 | 0.975 |

Table 10: Original Attacks with original and modified loss for comparison

Although the introduction of Reconstruction Loss did mitigate some of the attacks, it also diminished performance in some other attacks. Overall, this loss function can be improved further by calibrating the weights assigned to each loss better or even introducing a new loss term other than L1 loss we have shown.

| Attacker Name | Original | Modified |
|---|---|---|
| DiffWM Attacker | 0.933 | 0.900 |
| Black and White Attack | 0.913 | 0.975 |
| Lateral Inversion Attack | 0.960 | 0.950 |
| Sharpening Attack | 0.987 | 0.975 |
| Salt and Pepper Noise Attack | 0.833 | 0.775 |
| Hue Change Attack (0.3) | 0.773 | 0.775 |
| Elastic Deformation Attack | 0.967 | 0.950 |
| RGB to HSV Attack | 0.980 | 0.950 |
| Color Balance Attack | 0.993 | 1.000 |
| Gamma Correction Attack | 0.987 | 0.975 |
| Histogram Equalization Attack | 0.993 | 1.000 |
| Log Transform Attack | 1.000 | 1.000 |
| Color Jitter Attack | 0.960 | 0.950 |
| Color Quantization Attack | 0.807 | 0.900 |
| Sepia Attack | 0.900 | 0.950 |
| Posterization Attack | 0.960 | 0.975 |
| Anisotropic Diffusion Blur Attack | 0.647 | 0.675 |
| Directional Gaussian Blur Attack | 0.947 | 0.950 |

Table 11: Attacks proposed in our work with original and modified loss for comparison

## C    Adversarial Attack Analysis

### C.1    Composite Attack Results

Here we show the results for combinations of most successful attacks with adversarial attacks. All attacks are performed in standard configurations mentioned in Section A.1, an adversarial attack is carried out with a budget of 0.05 and 50 PGD steps.

### C.2    Attack Budget and Step Count Analysis

We analyze the interaction between perturbation budgets $\epsilon \in \{0.05, 0.1\}$ and PGD step counts $k \in \{5, 10, 20, 50\}$.

Although WDR did go down for higher budgets as expected , due to variations specifically in fourier space, the image quality suffers due to artifaction in the images. Thus the budget should be regulated closely as per the goals of the adversary. The variation of WDR in composite attacks is also almost in proportion to changes in the base case.

| Attack Name | WDR Score |
|---|---|
| Base | 0.7547 |
| diff_attacker_60 | 0.4906 |
| jpeg_attacker_50 | 0.7736 |
| brightness_0.5 | 0.7736 |
| Motion_blur | 0.5094 |
| contrast_0.5 | 0.7358 |
| vibrancy_1.25 | 0.7925 |
| black_white | 0.3962 |
| lateral_inversion | 0.6038 |
| Gaussian_blur | 0.7925 |
| AnisotropicDiffusion_blur | 0.3774 |
| DirectionalGaussian_blur | 0.4906 |
| sharpening | 0.7358 |
| salt_pepper_noise | 0.4151 |
| hue_change_0.5 | 0.0755 |
| posterization | 0.6981 |
| sepia | 0.4151 |
| rotate_90 | 0.0943 |
| rotate_180 | 0.0189 |
| rotate_270 | 0.1887 |
| lateral_rotate | 0.0189 |
| bm3d | 0.7736 |
| all | 0.0566 |

Table 12: WDR Scores for Various Attack Composition with Adversarial Attack

| Budget | Number of Steps | | | |
|---|---|---|---|---|
| | 5 | 10 | 20 | 50 |
| **0.05** | 0.8302 | 0.8278 | 0.8112 | 0.7546 |
| **0.1** | 0.6102 | 0.6032 | 0.5894 | 0.5206 |

Table 13: WDR with different combinations of perturbation budgets and number of PGD optimization steps.

