# OpenReview forum: "Reproducibility Study of "Attack-Resilient Image Watermarking Using Stable Diffusion""
_TMLR — Rejected by TMLR_

### Review · Reviewer_MJA4 · 2025-03-13

**Summary Of Contributions:**

This paper proposes a reproducibility study and robustness evaluation of "Attack-Resilient Image Watermarking Using Stable Diffusion" by Zhang et al. (2024), which introduces ZoDiac for attack-resilient watermarking. While replicating its >90% watermark detection rate (WDR) against diffusion-based attacks, the study uncovers vulnerabilities under adversarial and geometrically asymmetric attacks. Gradient-based adversarial perturbations, absent in prior evaluations, reduce WDR, while rotationally asymmetric attacks drop it below 65%. A proposed loss function improves robustness, but composite attacks still reduce WDR to near zero, exposing multi-stage vulnerabilities.

**Audience:**

Yes

**Claims And Evidence:**

Yes

**Requested Changes:**

Overall, this reproduction report is well-executed, providing detailed insights into the reproduction process and additional experiments. It also highlights some vulnerabilities of ZoDiac, which may be of interest to a subset of the TMLR audience, though its impact may be somewhat limited.

To strengthen the paper, I recommend the authors address Weakness 2 and Weakness 3 for a better work.

**Strengths And Weaknesses:**

### **Strengths**
1. The paper is generally well-written and easy to follow.
2. The experiments are extensive, and the description of the experimental settings is detailed and transparent.
3. The paper provides new insights into the vulnerabilities of the reproduced ZoDiac method, which makes for an interesting contribution.

### **Weaknesses**
1. **Limited significance**: Since ZoDiac's code is publicly available, its reproduction may have limited impact. A more valuable reproduction study would focus on high-impact methods that lack publicly available implementations.
2. **Effectiveness of the L1 explicit loss**: The proposed L1 explicit loss in Section 4.2.4 does not appear consistently beneficial based on the reported results. While it sometimes improves performance, it also fails in certain cases, limiting its overall significance. The authors should consider designing a more targeted loss function that directly addresses the vulnerabilities of ZoDiac identified in this study (e.g., rotational attacks).
3. **Grammatical and contextual errors**: The paper contains several grammatical mistakes and incomplete references. For example:
   - "The implementation of this experiment **is considers** WDR ..." (incorrect phrasing).
   - "(Results depicted in Table **??**)" (missing reference).

---

> ### Author Response · Authors · 2025-03-21
> **Response to Reviewer MJA4**
>
> We deeply appreciate the prompt review and we have addressed your concerns accordingly as follows:
>
> - Weakness 1:  The focus points of our reproducibility study are to validate and thus accept/reject the claims made by the authors of the original paper. To that end, our main focus has been towards stress testing the ZoDiac framework in order to properly establish the strengths and vulnerabilities of the method, requiring us to write a lot of the code from scratch. Additionally, the choice of the paper was not based on the availability of code, but rather on the fact that the authors claim to have better results than the previous SOTA frameworks(e.g. Stable Signature). Hence, while it is the case that the code is publicly available, we have conducted significant experiments that provide comprehensive analysis and fresh insights into the limitations of the method.
> - Weakness 2: We would like to emphasize that we do not present L1 loss specifically as a novel introduction, but an ablation introduced as a standard practice of using reconstruction loss in the watermarking research domain [1][2]. We would also like to highlight that authors have included this loss as an option in their example notebook but have neither discussed this in the original paper nor provided results regarding the same.  Therefore, we find it imperative to explore this augmentation in the reproducibility study. Furthermore, we have also included the need for a better reconstruction loss as a future direction to our work as finding a new loss function is outside the scope of our reproducibility study.
> - Weakness 3: We are grateful to the reviewer for pointing out the grammatical and contextual errors, we have amended the mistakes and have now proofread the paper thoroughly, hoping to have rectified any and all grammatical errors. Further, for additional clarity we have updated the descriptions of tables and figures.
>
> ### References
>
> [1]Jun Zhu, Yuedong Dong, Chengjun Li, Rui Long, Liang Zhang, and Weiqiang Huang. Hidden: Hiding data with deep networks. arXiv preprint arXiv:1804.00314, 2018.
>
> [2] Y Wan, Z Hu, and S Xie. Deep learning based invisible watermarking: A new framework. IEEE Access,
> 2019.

---

> > ### Comment · Reviewer_MJA4 · 2025-03-21
> >
> > Thanks for the response. I have no other questions.

---

### Review · Reviewer_jSyh · 2025-03-27

**Summary Of Contributions:**

This paper conducts a reproducibility study on "Attack-Resilient Image Water-marking Using Stable Diffusion" and its underlying method, ZoDiac. The authors first verify the core claims made in the original paper regarding ZoDiac. Based on their findings, they proceed to conduct extended experiments that examine the performance of ZoDiac under various types of attacks, including directional blurring, adversarial, rotational, and composite adversarial-geometric attacks. These experiments reveal vulnerabilities in ZoDiac's robustness to these attack types. In response, the authors propose modifying the ZoDiac loss function to address these concerns, but their results also highlight trade-offs introduced by this augmentation.

**Audience:**

Yes

**Claims And Evidence:**

Yes

**Requested Changes:**

I appreciate the authors' effort in conducting this reproducibility study. The paper not only verifies the claims made in the original paper but also reveals weaknesses, which is a valuable contribution to the field. However, I would have liked to see more context on why ZoDiac was chosen for evaluation.

**Strengths And Weaknesses:**

Strengths
    * The experimental evaluations in this paper are well-designed and effectively verify the primary claim made about ZoDiac.
    * The authors' reasoning for conducting extended experiments is clear, and their understanding of existing systems allows them to identify potential failure scenarios for ZoDiac.
    * The proposed augmented loss function offers partial mitigation of ZoDiac's weaknesses.

Weaknesses
    * A key limitation of this study is its focus on the white-box attack setting. To strengthen the findings, it would be beneficial to explore ZoDiac's performance in black-box or gray-box settings as well.
    * The proposed augmentation of the loss function only partially addresses ZoDiac's vulnerabilities and introduces trade-offs. A more effective solution with fewer compromises could improve the paper's impact.
    * In the introduction, while the authors provide a brief explanation for focusing on ZoDiac, further discussion on this topic would be beneficial to provide context and depth.

---

> ### Author Response · Authors · 2025-04-03
>
> We deeply appreciate the kind review. We have addressed the changes as requested. Our choice of the ZoDiac paper was based on the fact that the authors claim to have better results than the previous SOTA frameworks(e.g. Stegastamp,SSL, CIN, etc.) without significant computational overheads such as in Stable Signature. We have included the same in the introduction of our paper.
>
> We hope this addresses your concern.

---

### Review · Reviewer_BwbU · 2025-03-29

**Summary Of Contributions:**

The paper reproduces the paper "attack-resilient image watermarking using stable diffusion" from the github repository provided by the authors. The study confirms that the results presented in the original paper are correct. This is great and I believe that the independent verification of results is important for the trust in the science. The paper then shows that the watermark is not robust against different type of attacks than with respect to those it was tested in the paper. This is not that surprising.

**Audience:**

No

**Broader Impact Concerns:**

I do not have any concerns about the dangers.

**Claims And Evidence:**

Yes

**Requested Changes:**

Besides giving credit to original works in watermarking domain, I do not have request. I just find the results in the second part of the paper kind of obvious for people knowledgeable about watermarking technology.

**Strengths And Weaknesses:**

The first part of the paper is reproducing the original paper and confirming the outcomes.

The second part of the paper goes into additional study with respect to which distortions the watermarking algorithm is robust and to which it does not. It is here where the paper sort of fall short. First of all, I am surprised, why authors are so much concerned with rotation invariance and they do not consider clipping and resizing. These two distortions are very common image manipulations, way more common than rotation.

From reading I have an impression that authors neglect the art done in watermarking not based on deep learning. Fabien Petitcolas was very active in evaluating robustness of watermarking schemes, e.g. [1] yet he is not even mentioned, despite his work has 1000 citations. Similarly, authors talks about adversarial attacks by knowledgeable attacker. They were known to watermarking community [2,3] well before they become fashionable within the field of deep learning.

According to my limited knowledge of watermarking I think it is futile to search for a watermarking system robust with respect to all possible distortion. The problem is effectively open-ended (as many security problems), because there are many different methods how to distort the image. In practice, the watermarks are designed to be robust against some set of attacks and distortions. With respect to this, it is not surprising that there exists attacks which removes the watermark.

[1] Kutter, Martin, and Fabien AP Petitcolas. "Fair benchmark for image watermarking systems." Security and watermarking of multimedia contents. Vol. 3657. SPIE, 1999.technology: coding and computing. IEEE, 2001. [2] Comesana, Pedro, Luis Pérez-Freire, and Fernando Pérez-González. "Blind newton sensitivity attack." IEE Proceedings-Information Security 153.3 (2006): 115-125. [3] El Choubassi, Maha, and Pierre Moulin. "New sensitivity analysis attack." Security, steganography, and watermarking of multimedia contents VII. Vol. 5681. SPIE, 2005.

---

> ### Author Response · Authors · 2025-04-03
> **Response to Reviewer BwbU**
>
> We appreciate the concerns raised by the reviewer and have tried to address them to the best possible:
> - We have addressed the Rotation and Lateral Inversion Attacks with higher emphasis due to the fact that the ZoDiac embeds a rotationally symmetric watermark which is already established to be more resilient towards geometric attacks[1][2]. The original paper has also focused on Rotational Attacks which represents no loss of information. We have extended the results to show that near zero WDR can be achieved based on purely Rotational Attacks. As also pointed out by the reviewer that watermarks are designed to be robust against some set of attacks and distortions, we have evaluated ZoDiac on the attacks which were included in the original paper or close extensions to them.
>
> - We appreciate the reviewer for pointing this out and sincerely apologise for missing the citations to these works. We have now fixed the issue and included them in the paper.
>
> - We strongly agree with the reviewer that a method is robust against only a set of attacks and distortions. It is for this very reason that it becomes even more imperative to know the exact set of attack paradigms under which the watermark methodology performs poorly or unsatisfactorily. This is also important from a practical implementation viewpoint so as to be aware of the exact set of conditions under which the methodology might fail. Thus, through our reproducibility study, we have stress tested the ZoDiac framework under relevant attack paradigms.
>
> We hope this addresses all you concerns.
>
> ## References
> [1]Vassilios Solachidis and Loannis Pitas. Circularly symmetric watermark embedding in 2-d dft domain.
> IEEE transactions on image processing, 10(11):1741–1753, 2001
>
> [2]Yuxin Wen, John Kirchenbauer, Jonas Geiping, and Tom Goldstein. Tree-rings watermarks: Invisible
> fingerprints for diffusion images. In Thirty-seventh Conference on Neural Information Processing Systems,
> 2023.

---

> > ### Comment · Reviewer_BwbU · 2025-04-04
> >
> > So what was the criteria for selecting the attacks? Because the original Stirmark test suit, which you kindly not cited, actually copied the type of distortions media would undergo at that time. Therefore if you are interested in limits of the watermark, which was not stated in the paper", you need to develop a test suite covering set of contemporary common image manipulations. The choice of the attacks seems ad-hoc to me. Was there any methodology behind the selection?

---

> > > ### Author Response · Authors · 2025-04-05
> > >
> > > We have extended the attacks under domains under which ZoDiac was being tested , providing a more transparent picture of it's utility under those attack paradigms. Our choice of attacks was based on the following factors - extending the attacks presented in the original paper and some of the known limitations of the SSIM score used in ZoDiac's loss function. Here is a brief overview of the motivation behind the choice of attacks:
> > > - Directional Blurring Attacks: These set of attacks were chosen to disturb the rotational symmetry of the watermarking algorithm. These were specific extensions of blurring attack i.e. G-Blur attack demonstrated in the paper.
> > > - Changing Color Hue: These attacks were based on the known limitation of the SSIM metric which fails to capture the color distortions.
> > > - Rotational and Geometric Attacks: These attacks were chosen to quantify the WDR under variation of rotation attack already discussed in paper.
> > >
> > > A detailed explanation to each of these attack choices are already included in the paper.
> > >
> > > We have not tested using the Stirmark test suite because it is an outdated benchmark, missing out on important attacks that are relevant to current settings, e.g. AI based attacks (Diffusion/GANs based reconstruction and editing attacks etc.). Also, the attack set that we used was relevant in the context of ZoDiac which was the goal of this study rather than developing a generalised test suite applicable to multiple watermarking frameworks.
> > >
> > > While we haven't developed a full benchmark, we have ensured to provide meaningful extensions to the attacks tested in the original paper, which directly aligns with the scope of the reproducibility study. Development of a full benchmark can be considered as a subsequent work, and outside the scope of a reproducibility study.

---

### Decision · Action_Editor_hMJ5 · 2025-05-25

**Recommendation:** Reject

**Comment:**

This paper presents a reproducibility study of "Attack Resilient Image Watermarking Using Stable Diffusion" (ZoDiac), confirming its core claims while extending the evaluation to adversarial and asymmetric attack settings. The study demonstrates that ZoDiac is vulnerable to attacks not originally tested, which can be a bit interesting to some of the audience.

The reviewers are divided, with positive evaluations acknowledging the value of verifying ZoDiac's performance and uncovering overlooked limitations, while the more critical perspective emphasizes that the work lacks historical context and methodological rigor in its robustness claims. The authors have made a genuine effort to address concerns, including justification of attack choices and clarifying their scope as a focused reproducibility study.

After re-reading the paper, it remains unclear what the main claims of this reproducibility study are, and the authors are encouraged to state them explicitly. To substantiate the assertion made in the abstract---"we identify critical vulnerabilities under adversarial and geometrically asymmetric attack paradigms"---the authors must more thoroughly engage with prior work on the systematic evaluation of watermarking robustness. This includes situating their findings within the broader historical context and clarifying whether any of the identified vulnerabilities are already known to the field. Without this engagement, the current version lacks the necessary foundation to support its claims and the contribution risks appearing incremental or redundant, which in term decreases the interest of TMLR audience.

**Audience:**

some, but very few (see meta-review)

**Claims And Evidence:**

no (see meta-review)

**Resubmission Of Major Revision:**

The authors may consider submitting a major revision at a later time.